# "[A]n Exterior Air of Pilgrimage": The Resilience of Pilgrimage Ecopoetics and Slow Travel from Chaucer's *The Canterbury Tales* to Jack Kerouac's *On the Road*

Susan Signe Morrison

Department of English, Texas State University, 601 University Dr., San Marcos, TX 78666, USA; morrison@txstate.edu

**Abstract:** While the Beats can be seen as critical actors in the environmental humanities, their works should be seen over the longue durée. They are not only an origin, but are also recipients, of an environmentally aware tradition. With Geoffrey Chaucer and Jack Kerouac, we see how a contemporary American icon functions as a text parallel to something generally seen as discrete and past, an instance of the modern embracing, interpreting, and appropriating the medieval. I argue that *The Canterbury Tales* by Chaucer influenced Kerouac's shaping of *On the Road*. In the unpublished autograph manuscript travel diary dating from 1948–1949 (*On the Road* notebook), Kerouac imagines the novel as a quest tale, thinking of pilgrimage during its gestation. Further, Kerouac explicitly cites Chaucer. His novel can be seen not only in the tradition of Chaucer, but can bring out aspects of pilgrimage ecopoetics in general. These connections include structural elements, the spiritual development of the narrator, reliance on vernacular dialect, acute environmental awareness, and slow travel. Chaucer's influence on Kerouac highlights how certain elements characteristic of pilgrimage literature persist well into the modern period, in a resilience of form, language, and ecological sensibility.

**Keywords:** pilgrimage; Geoffrey Chaucer; *The Canterbury Tales*; Jack Kerouac; *On the Road*; ecocriticism; ecopoetics; slow travel; vernacular

---

Artfully citing the poet and translator Peter Stambler, Hong Cheng refers to the intertextual links among seemingly disparate authors, time periods, and literary works as "'*encounters*, perhaps conversations'" (Cheng 2006, p. 135; Stambler 1996, p. 13). These metaphors—encounters and conversations—suggest how influence is not simply derived one way in a forward thrusting chronology. A contemporary interlocutor, by encountering and conversing with a past actor, can be seen as changing or reforming that initial inspiration for the revelation of present-day viewers, readers, and witnesses. Shakespeare, for example, adapted the long tragic romance, *Troilus and Criseyde*, of Geoffrey Chaucer to craft his own *Troilus and Cressida* for an early seventeenth-century audience. James Joyce used Homer's *Odyssey* as a blueprint for his protagonist in early twentieth-century Dublin. Margaret Atwood's *The Penelopiad* likewise used Homer to very different ends than Joyce as she concocted a feminist indictment of the 'hero'—or is he a villain?—Odysseus. All of these encounters or conversations expand, question, and morph the earlier works in which they are rooted.

The Beats are shaped by previous writers even as they stake new claims on the literary landscape. Critical actors in the environmental humanities, the Beats should be seen over the longue durée as the recipients of an environmentally aware tradition, ranging from the Tang dynasty poet Han-shan to—as this essay focuses on—the fourteenth-century English poet, Geoffrey Chaucer. Kerouac's

Anglo-American modernist predecessor, T. S. Eliot, likewise influenced by Asian cultural references, proves instructive here. In "Tradition and the Individual Talent", Eliot points out how

> the historical sense compels a man to write not merely with his own generation in his bones, but with a feeling that the whole of the literature of Europe from Homer and within it the whole of the literature of his own country has a simultaneous existence and composes a simultaneous order. This historical sense [ … ] is what makes a writer traditional. And it is at the same time what makes a writer most acutely conscious of his place in time, of his own contemporaneity. (see Eliot 1919/2020)

While conceptualizing *On the Road*, Jack Kerouac references *The Canterbury Tales* by Chaucer. The parallels between the two authors expose how elements characteristic of medieval pilgrimage literature resiliently persist into the twentieth century. Yet, Kerouac's "historical sense" never trumps his own sense of "contemporaneity". Nurtured by past tradition, Kerouac seeds it with his own innovations for future writers.

While medievalists and Beat scholars rarely converse with one another, such an encounter proves a catalyst for exposing: (1) elements characteristic of pilgrimage literature in general; (2) how an earlier writer surprisingly shapes an unexpected protégé; and (3) the ecocritical aspects of two major works of literary history which are seldom yoked. Recent scholarship in Chaucer studies (see Morrison 2019) explores the ecopoetics of medieval literary pilgrimage. By introducing these insights to Kerouac's novel—as we will see, overtly connected to Chaucer—the ecocritical insights of *On the Road* can be made more readily evident. Both works emerge from pilgrimage ecopoetics, exploring vernacular language, linguistic and physical contingency, and slow walking, where slowness functions as a form of rebellion (see Arnds 2020, p. 2). This resilience of certain literary, religious, and ecological conventions links the medieval with the Beat, the European and Asian with the American, and the physical with the spiritual. Such "'*encounters*, perhaps conversations'" (see above) can then be recognized as expanding, as though through concentric circles, the associations among vastly—only apparently unrelated—modes of writing and thought. This indicates a sophisticated reception by Kerouac of long-established tropes and traditions in literature, religion, and ecological thought and awareness. After establishing clear referential and structural links between Chaucer and Kerouac, this essay looks briefly at an Asian counterpart to highlight aspects of the Anglophone works, before exploring faith, spiritualized landscapes, vernacularity, topopoetics, and the role of slowness in the ecologically-conscious pilgrimage journey.

## 1. The Literary Tradition of Pilgrimage Literature: Finding the True Path

Pilgrimage as a ritual process expresses the spiritual yearning, interior journey, and physical enactment of the pilgrim actor. Literary works utilizing pilgrimage as a formal element—whether structurally or thematically—are rooted in the metaphor of the road. The central soul of such a work, frequently narrating his or her own literal and figurative pilgrimage, makes mistakes which ultimately guide the individual to God. This section of the essay explores aspects of Kerouac's faith as expressed in *On the Road*, linking that novel to a clear predecessor: Chaucer's *The Canterbury Tales*. Both works, structured around pilgrimage, explore false paths on the spiritual journey.

Chaucer, a product of fourteenth-century England, was steeped in Christianity. Late medieval Catholic faith espoused the necessity of pilgrimage, an interior and exteriorized journey to expiate sins and gain absolution. While Kerouac lived hundreds of years later, his religious upbringing, much discussed by his biographers, can be seen as naturally and inevitably leading him to be open to the tradition and concept of Christian pilgrimage. Numerous critics have explored his French-Canadian Catholic ethnicity, with Richard Sorrell arguing that Kerouac took the "'terrible holy majesty' of Roman Catholicism more seriously than most of his childhood peers" (Sorrell 1982, p. 191; also, Christy 1998, p. 87; Charters 1973, p. 199). He describes himself as being "not 'beat' but strange solitary crazy Catholic mystic [sic]" (Kerouac 1960, p. vi; also quoted in Charters, The Portable Jack

Kerouac, Kerouac 1995b, p. xxv). Writing that "[b]eat doesn't mean tired, or bushed, so much as it means *beato*, the Italian for beatific," Kerouac claims that he wants to "'be in a state of beatitude, like St. Francis'" (Amburn 1998, p. 205; see Christy 1998, pp. 94, 40, who argues Kerouac most closely resembles St. Augustine of Hippo). But does this Christian dimension alone to his writing mean that this famous novel is a pilgrimage work? Jim Christy argues that *On the Road* is a "religious parable, a holy search for meaning, a supercharged *Pilgrim's Progress*" (Christy 1998, p. 26). Certainly, moments of spirituality infuse Kerouac's prose. Concerning the mambo music in the Gregoria whorehouse in Mexico, Sal hears trumpets "like the sounds you expect to hear on the last day of the world and the Second Coming" (286; all Kerouac textual allusions, unless otherwise noted, refer to Kerouac 1991).

Rather than mere inference, there exists direct evidence for pilgrimage being on Kerouac's mind. The Harry Ransom Center at the University of Texas at Austin has Kerouac's unpublished autograph manuscript travel diary dating from 1948–1949, when he was writing the first stages of *On the Road*. Here, Kerouac imagines his work as a quest tale, explicitly thinking of pilgrimage. The narrator would play a kind of Sancho Panza figure to the main character (Charters 1991, p. xv). Concerning this version of the novel, Kerouac writes the following:

> The hero is a man in his late 20's who has lived a lot, and who ends up in a jail, thinking, finally, that he needs to "seek an inheritance incorruptible, undefiled, and that fadeth not away," in the words of [John] Bunyan. (Kerouac 1948–1949, 25 March 1949; see Christy 1998, p. 26)

Explicitly invoking the seventeenth-century Protestant work, *The Pilgrim's Progress* (1678), Kerouac ponders pilgrimage. Four days after this journal entry, he writes, "*Pilgrimage* [ . . . ]. My interest in the 'beat': it must be because they're not only poor, but *homeless* [ . . . ]. Their lives have an exterior air of pilgrimage (wandering + impoverished) [ . . . ]" (On the Road notebook, Tues. 29 March 1949).

Three months after these notebook entries, Kerouac writes a letter to Elbert Lenrow, from whom he took a twentieth-century American novel class at the New School in 1948. This letter, from 28 June 1949, includes fascinating tidbits, including how Kerouac picked up a copy of Edmund Spenser's complete poems for fifty cents. Kerouac adds that he was depressed by Boethius's life until he read *The Consolation of Philosophy*, a key work for Chaucer, who translated it into Middle English. In Kerouac's letter, the following passage appears: "In Chaucer, by the way, 'bone' is PRAYER" (Selected Letters, Kerouac 1995a, p. 207). This letter suggests Kerouac clearly understood some Middle English, where the word *bōn* (also *bone*, *boin*(e) *bōnen* and *bones*), signifies *act of prayer*, *petition*, *request*, or *boon* [see McSparran and Schaffner 2000–2018, entry *bōn* n.(2)]. Kerouac must have read Chaucer at some point, possibly exposed to it by his friend William Burroughs, who enjoyed classes on both Chaucer and Shakespeare at Harvard (Schumacher 1992, p. 31).

While seemingly wildly disparate—poetry versus prose, medieval versus modern, pre-industrial versus technologically advanced—both *The Canterbury Tales* and *On the Road* mirror one another in a number of structural ways: both are first-person literary works with a narrator standing in for the author. This is typical of pilgrimage literature in general. A first-person, frequently flawed, individual allows the reader to identify with the speaker and enter into the pilgrimage with the writer-guide-companion. In Chaucer's poem, the narrator is actually named Geoffrey Chaucer, though a bumbling and naïve variation of that sophisticated and revolutionary poet. Sal—reminiscent of Salvatore or Savior—Paradise, whose name alone evokes the pilgrimage author Dante, twice describes his journeys as a "pilgrimage" (139, 303). The third sentence of the novel announces " . . . the part of my life you could call my life on the road" (1). Dean is ideal for this expedition, seeing as he was "born on the road" (1). Simply being "on the road" signals the pilgrimage theme. "And this was really the way that my whole road experience began" (7). Sal calls it "the holy road" in conjunction with "pilgrimage" (139). When Sal tells us "the road is life" (212), this fits into the long Christian allegorical tradition of understanding life itself as a pilgrimage.

And not just Christian traditions. As Cheng points out, twentieth- and twenty-first-century American nature writers, such as Gary Snyder and Jack Kerouac in *The Dharma Bums*, draw on ancient

Chinese lore for inspiration to express an echoing "cultural orientation" (Cheng 2006, p. 136). Kerouac repeatedly invokes the ancient Chinese poet "Han Shan" in *The Dharma Bums* (see Cheng 2006, p. 135; also Kerouac 2006, pp. 13–16, as well as later in the novel). While Han-shan dates from a period and geography far distant from Kerouac and even Chaucer, certain aspects of his influential interaction with nature and landscape parallels that of Sal's in *On the Road*. Yes, the technology is vastly different. Sal and Dean travel quickly by means of an automobile. Nevertheless, Kerouac remains influenced by slower modes of understanding the self, as seen in earlier works with pilgrimage as a modifying metaphor.

Asian spirituality and European medieval Catholicism deviate in multiple ways, most clearly in divergent religious cultures. Yet, both play out the metaphor of soul seeking on the literal plane—through pilgrimage, a physical journey which contributes to and sometimes even distracts from the inner journey to self-knowledge, peace, and a deepened faith. Pilgrimage works contain paths which do not lead to pure spirituality, in order, ultimately, to suggest a better way. The pilgrim cannot recognize the correct path to righteousness or enlightenment without first meandering falsely. In Dante's *Inferno*, the pilgrim-narrator is not meant to stay in hell, merely witness and learn from it. The fifteenth-century English pilgrim, Margery Kempe, begins her life story by confessing her sins of lust, pride, and envy before relaying her many visions of Christ. Chaucer and Kerouac integrate counterexamples, dead ends which signal how *not* to act. Going astray literally and metaphorically actually feeds an environmentally aware response to one's surroundings. Straying leads to surprises and disruptions, preventing the pilgrim-actor from feeling as though he can master the world around him.

The pilgrim-narrator establishes his identity through contrast with an unconventional figure—the self-professed conman, as embodied by Chaucer's Pardoner and Kerouac's Dean—to forge his own sense of self on his spiritual journey. The Pardoner, a seller of indulgences, conveys an indeterminate morality. In awe, the innocent narrator—just after he has described how false and conniving the Pardoner actually is—calls the cheating religious actor a "noble ecclesiaste" (I.708 "noble ecclesiast"; all Middle English passages from Chaucer 1987; modern English translations adapted from Chaucer 2019). Similarly, Sal calls his companion Dean "noble", even though his behavior has been far from admirable. Sal calls Dean a "con man"; he was "conning me" (4). Sal naively sees Dean as a conduit to higher spiritual truths, much as Geoffrey mistakes the chiseling Pardoner for a worthy churchman. Both the Pardoner and Dean brag to their listeners about being cheaters before they try to swindle their audience. Chaucer's Pardoner tells the pilgrims in great detail how he dupes his victims, then attempts to get money from those he has just confessed to. Similarly, when a man picks Sal and Dean up in his car, they stop at a hotel: "Dean tried everything in the book to get money from the fag.... Warning him first that he had once been a hustler in his youth, Dean asked him how much money he had" (210). After they have a fight, Sal refers to Dean as the "holy con-man" (214; also 15) (Charters 1973, p. 81; also, Amburn 1998, pp. 353, 105, and Sorrell 1982, p. 194)—exactly what the Pardoner is.

Great talkers, these 'holy con men' are at last stifled when harshly spoken to. After the drunken Pardoner delivers a sermon against gluttony and drinking, the Host threatens to cut off the corrupt churchman's balls. Galatea aggressively deflates Dean with a verbal attack (193), rendering him verbally impotent and reduced to silence, much as the Pardoner is.

> This Pardoner answerde nat a word;
> So wroth he was, no word ne wolde he seye (VI.956–57).
> [This Pardoner answered not a word;/So angry he was, no word would he say.]

After Galatea lays into Dean, Sal reflects, "Then a complete silence fell over everybody; where once Dean would have talked his way out, he now fell silent himself . . . " (195). Exposed for what they are—con men whose power emerges from speech—they are denied a voice. Once the false guide is silenced, the pilgrim-narrator can explore true sociality on the right path.

## 2. Vernacularity, Sociality, and Topopoetics: Shaping the Pilgrim's Soulscape

Pilgrimage poetics reflect the ever-evolving state of the land imprinted by physical pilgrimage. Each writer's vernacular is location-dependent. Tim William Machan, in his exploration of Middle English, carves out a sociolinguistic model to argue for the vernacular language as an ecology (Machan 2003, p. 9). The metaphor of ecology suggests that the living, dynamic vernacular, integral to the ecopoetical aesthetic, constitutes a fertile means for understanding a specific place.

A trait of pilgrimage literature even in the Middle Ages, linguistic vigor reflects the forward propulsion of physical pilgrimage and its topopoetics, the language which springs naturally forth to express a specific and ever-evolving landscape and culture. Both Chaucer and Kerouac write using highly-charged, dynamic, and constantly morphing languages. Writing at the end of the fourteenth century, Chaucer profits from the recent influx of over ten thousand French words into the English language in the wake of the Norman Conquest three centuries earlier. He innovates new usages of words and even coins vocabulary. Kerouac's linguistic energy derives from a globalized English buzzing with slang and colloquialisms. In both texts, dead language inculcates inert spirituality and topographical doom, while linguistic liveliness both reflects and shapes the dynamism of the landscape and environment. Vibrant language signals and sparks a lively awareness of the world replete with both human and non-human actors.

Chaucer and Kerouac avow vernacular authenticity. In works where he emphasizes his role as a 'mere' translator or compiler, Chaucer's narrator emphasizes his powerlessness and lack of free will. He apologies for any rude material he must include in the name of accuracy—in his utterly fictional world. Chaucer's narrator, claiming he is honest, apologizes to the reader for his possibly offensive language.

> But first I pray yow, of youre curteisye,
> That ye n'arette it nat my vileynye,
> Thogh that I pleynly speke in this mateere,
> To telle yow hir wordes and hir cheere,
> Ne thogh I speke hir wordes proprely ... (I.725–729).
> [But first I pray yow, of your courtesy,/ That you do not attribute it to my rudeness,/ Though I speak plainly in this matter,/ To tell you their words and their behavior,/ Nor though I speak their words accurately.]

Like Geoffrey, Sal says his readers may have a hard time accepting the truth of what he writes. "And this was really the way that my whole road experience began, and the things that were to come are too fantastic not to tell" (7). This excuse of merely telling the truth by copying what he has heard echoes comparable sentiments in Chaucer's *Miller's Prologue*. There, the narrator makes sure that his readers know it is not his own personal evil intent for speaking so crudely, rather his aim is to be honest (I.3172–3181). He begs his reader not to chastise him and to choose another tale if preferred. In fact, he prays, "For Goddes love" (I.3171–2; "For God's love"), that the reader not condemn him for his apparent honesty. In Kerouac's work, his evident transparency could be rooted in his Catholicism. His faith is said to have influenced his poetics, with the supposed authenticity of spontaneous prose resembling unfettered prayer and confession. The effect of spontaneity was to permit the author to make associational leaps between languages and enhance wordplay (Kerouac 1998, pp. 69–70). Kerouac was a "man who knelt in prayer before writing and who acknowledged that the sacrament of confession had inspired his 'spontaneous prose' style of composition (it was sinful to hold anything back in the confessional—you purified yourself by telling all in an unrevised, unconstrained rush of words) [...]" (see Sorrell 1982, pp. 195–96). Linguistic contingency constitutes a tool for the pilgrimage writer. Though such protestations legitimate their so-called 'accuracy', both authors have command of what they are doing, perfectly capable of revising and editing as they wish.

Kerouac's iconic novel was influenced both through contingent encounters with authors such as Chaucer and edited by design. While amendment reflects the spiritual change to be undertaken by both literary pilgrim and the pilgrim-reader, the pilgrimage text itself is ever-evolving, obsessively

walked over (metaphorically) by the pilgrim writer (see Morrison 2019, p. 53). In terms of revision of the written text, medieval pilgrimage authors deal with amendment on the *literary* level. *The Canterbury Tales* itself exists in variant fragments. In spring of 2001, the famous scroll draft of *On the Road*, which Kerouac claimed to have completed in three weeks (thereby evoking Truman Capote's famous comment, "That isn't writing; it's typing" (Charters 1991, p. viii), was auctioned at Christie's in Manhattan for 2.4 million dollars (Shattuck 2001). The almost one-hundred-and-twenty-foot scroll consists of sections of paper about twelve feet long pasted together, with the seams reinforced by tape. The mythology that Kerouac wrote *On the Road* in a coffee and Benzedrine frenzy has been questioned. Despite the seductive legend of a spontaneously produced text, Kerouac amended his work considerably, from its initial conception in 1948 until its publication in 1957. The scroll shows Kerouac's editing—"words are changed, punctuation added, paragraphs indicated and entire passages crossed out in pencil and red crayon" (Shattuck 2001)—suggesting that he was a much more careful writer than legend suggests.

Both authors endorse the vernacular speech of their contemporaries by pleading for indulgences—those of the reader for their honest, truthful replication of 'real speech'. This creative use of the vernacular links Kerouac to the pilgrimage literary tradition. Medieval pilgrimage poems acknowledge the importance of using the vernacular on the grounds that such writing could change not only the poet's soul, but also that of the reader. The vernacular, necessary to affect inward spiritual transformation for (potentially) everyone, has a salvational aspect to it. Chaucer's poem, which includes parodies of Northern dialect in the *Reeve's Tale*, provides a map for vernacular language use in late fourteenth-century England, just as Kerouac's prose replicates the jaunty rhythms of mid-20th-century American patter. This vernacularity explodes with multiple unusual terms, from "jalopy" (1), "cold-water pad" (1), "chick" (2), "lam" (3), "benny" (3), "jargon" (4), "dingledodies" (5), "balled the jack" (14), "hobo" (26), *"Pisscall"* (29), "slaphappy" (32), "s'danged" (36), "sonumbitch" (43), "dangle" (43), "you fine gone daddy you" (43), "crazy cat" (48), "jazz American" (61), "beatest", "tea, weed, I mean marijuana", "bop", "boogie-woogie" (87), "hincty" (87), to "'Right-orooni'" (176). One odd parallel between the two writers concerns the influence of French. Chaucer's language reflects the recent incorporation of thousands of Norman French words, a typical sign of late medieval English. Kerouac's first language was French and he understood no English until he was six. Christy suggests that "Kerouac's acute sensitivity to the English language derived from his hearing it as an exotic tongue" (Christy 1998, p. 79; also see Charters 1991, p. xxv; Weinreich 1995, p. 40; and Giamo 2000, p. xv).

David J. Alworth argues that "Kerouac turned to the open road in order to reimagine sociality" (Alworth 2016, p. 82). True sociality requires the glue of the vernacular. Sal associates specific places and their names with the community they invoke: "And I tried to tell him what North Platte meant to me, buying the whiskey with the boys, and [Remi] slapped me on my back and said I was the funniest man in the world" (64). Kerouac increases this sociality through associating places with literary, filmic, and pop cultural references the reader can connect with. These include William Saroyan (81), *Sullivan's Travels* (83), *"Of Mice and Men*, with Burgess Meredith" (91); "I had a book with me I stole from a Hollywood stall, '*Le Grand Meaulnes*' by Alain-Fournier" (103); Hart Crane (119); W. C. Fields (121, and mentioned four times overall); "Louis-Ferdinand Céline" (137); Shakespeare (144), Groucho Marx (154); "[I]t made you think of Sam Spade" (170); "Eugene Sue's *Mysteries of Paris*" (192); and Proust (304). Dean is likened to a "mad Ahab at the wheel" (235). Such allusions suture the reader into the scene of Sal's world, making us privy to his social sphere.

Vernacularity is not just a linguistic term but can also refer to material practices. Rob Nixon has argued that "imposed official landscapes typically discount spiritualized vernacular landscapes" (Nixon 2011, p. 13), ones which contingently develop. An example of this tension between vernacular and official landscapes can be seen in Kerouac's novel. Hitchhiking suggests spontaneous movement and holy Mexico constitutes the vernacular landscape. Unplanned meandering, a more instinctual, even 'natural', response to terrain and built environment, allows for fortuitous meetings. While

pilgrimage seems as though it would be highly designed, it remains deeply wedded to the accidental, both on the literal and literary levels (Morrison 2019, p. 44). In actuality, pilgrims not infrequently strayed off the official path. Even in Chaucer's seemingly designed set-up, characters spontaneously arrive and disappear, such as the Canon and Canon's Yeoman. Early on while hitchhiking, Sal has already taken part in "five scattered rides" (10) (though see Müller 2016, p. 596, on how chance encounters diminish as literature increasingly becomes modern). Unlike slow medieval pilgrimage undertaken by walking or as an assemblage on horseback, Sal's pilgrimage is fueled by automotive mechanics and rhizomatic hitchhiking (especially Sections 2–4). The carnival owner in Iowa asks, "'You boys going to get somewhere, or just going?' We didn't understand his question, and it was a damned good question" (21). Unlike his medieval European counterparts who journey with purpose to specific holy shrines (Rome, the Holy Land, Santiago de Compostela) for healing, absolution, and the remission of sins, Sal apparently wanders in a propulsive randomness. Unlike his nineteenth-century urban counterparts, Sal acts as a flâneur of the rural landscape, a kind of "environmental phenomenologist" (Gersdorf 2013, p. 44).

With medieval pilgrimage as a concept helping shape *On the Road*, we recognize the outline of intention with the trip to Mexico. The country to the south symbolizes the ultimate pilgrimage site, due to its profound Catholicism infused with indigenous culture. "Behind us lay the whole of America and everything Dean and I had previously known: about life, and life on the road. We had finally found the magic land at the end of the road" (276). Mexican characters or Mexico itself carry what Ben Giamo calls "biblical magic" (Giamo 2000, p. xv, 21; also Amburn 1998, p. 215). Early on in the novel, Terry, Sal's lover, a Madonna figure, pitiful and childlike, functions as his redemptress, an intercessor reminiscent of the Virgin Mary, a figure Kerouac's brother Gerard claimed to have seen in a vision before he died at the age of nine. After she feeds Sal, "[i]t was Terry who brought my soul back" (97). The intersection between place and soul is most pronounced in Mexico or in conjunction with Terry, whose Mexican origin poignantly calls to Sal. She prefigures Sal's ultimate understanding that Mexico will provide sacred wisdom.

Nature writers, overcome by modern demands on time and psyche, often retreat to wilderness spheres to seek out "the solitude of mind totally free of the noisy material world" (Cheng 2006, p. 138). They escape from so-called civilization, which all too often proves to be detrimental or chaotic, to a *higher* form of living in less-populated places. For Sal, that sphere is Mexico, set up in the novel as—admittedly problematically—a 'primitive', exoticized, and less technologically fraught zone of freedom. However politically suspect the use of Mexico may be in Kerouac's hands, it spurs spiritual awakening, where "landscape and soulscape" (Cheng 2006, p. 138) are one. Wilderness "suggests something like freedom . . . to contemporary American nature writers, [becoming] a symbol of civilization in a higher form and a course for deep thoughts and inspiration" (Cheng 2006, p. 141). For Kerouac's protagonist, connections to Mexico personally or geographically catalyze emotional liberation. "It was only Nuevo Laredo but it looked like Holy Lhasa to us" (274). Sal's "soulscape" (Cheng 2006, p. 138) finds peace in Terry's world and culture, one he yearns to be part of.

Just as Chaucer cites natural and built environmental sites to place us geographically in a specific place, Kerouac's linguistic tapestry beautifully evokes the landscape. Concerning New Mexico, Kerouac writes, "We were in the mountains: there was a heaven of sunrise, cool purple airs, red mountainsides, emerald pastures in valleys, dew, and transmuting clouds of gold; on the ground gopher holes, cactus, mesquite" (164). In an ecocritical reading of *On the Road*, we see that in Kerouac's language, toponyms—specific linguistic identifying markers—matter. Sal: "'I love boxcars and I love to read the names on them like Missouri Pacific, Great Northern, Rock Island Line'" (41). A topopoetics is rooted in the vernacular of place through dialectical idiosyncrasies and a precise and local lexicon. As it modifies, the vernacular is uniquely suited to express the physical, environmental, and spiritual dimensions of pilgrimage (Morrison 2019, pp. 47, 50). Kerouac references biologically indigenous plant-life, like "those lovely California cottonwoods and eucalypti" (78) and "yucca cactus and organpipe" (276). For Sal, the environment becomes a literary work for him both to inscribe upon and interpret. In Louisiana,

he reflects, "This was a manuscript of the night we couldn't read" (157). "[T]he sheer contingency of the natural world" prevents us from feeling as though we can master the world around us (de Luca 2016, p. 219). Contingency allows for a measured appreciation of American topography, vernacular accents, and geography.

### 3. Slow Travel and Environmental Insight

Technologically, Chaucer's pilgrims would be slower than many a present-day pilgrim who speeds to Rome, Jerusalem, or Mecca by plane. Chaucer's poem emerges out the walking tradition of pilgrimage, which "involves slow organic movement through a landscape" (Northcott 2008, p. 215). Our first encounter with Chaucer's Wife of Bath emphasizes how she practiced "wandrynge by the weye" (*The Canterbury Tales* I.467; "wandering by the way"). Wandering here suggests dalliance, both sexual and in terms of tarrying, slowing down to revel in the present moment. Slow travel infuses his characters' actions. The narrator of "Tale of Sir Thopas" takes the time to mark and identify each little aromatic plant, from the "lycorys and the cetewale,/ And many a clowe-gylofre,/ And notemuge to putte in ale" (VII.761–63; "licorice and the zedoary,/ And many a clove-gillyflower;/ And nutmeg to put in ale"). Slowness allows time to be intimate withone's surroundings.

While Kerouac's *On the Road* has been called an "egocentric ... speed trip", I would like to emphasize how it prefigures "eco-centric slow travel". Examples of slow travel famously include William Least Heat-Moon's personal journey on backroads of rural America in his best-selling *Blue Highways* (Arnds 2020, p. 4) and the reflections of Rebecca Solnit in her book *Wanderlust: A History of Walking*. Walking, an inherently slow medium, permits the actor unexpected revelations: "[t]he random, the unscreened, allows you to find what you don't know you are looking for, and you don't know a place until it surprises you" (Solnit 2000, p. 11). Scholars in the Environmental Humanities have looked at slowness as key to various ecocritical maneuvers criticizing the swift pace of technological globalization. These devices include "[s]low [s]cholarship for the [a]nthropocene" (Bergthaller et al. 2014), promulgated by eco-critics as "an ideal ... which cultivates thinking across different spatiotemporal scales" (Bergthaller et al. 2014, p. 261). Dramatically slow art projects—such as Katie Paterson's *Future Library* project, inaugurated in 2014—are layered with the ideals of slowness and patience as ecocritical strategies. In Paterson's piece, for example, renowned writers and poets have pledged to contribute works which will remain unpublished until 2114. An ongoing concert for 639 years in Germany had its first sound change since 2013 in 2020. This concert of John Cage's organ recital began in 2001 and will continue until 2640 (Hickley 2020). Such slowness urges us to pay attention to the present, while suggesting hope in the future.

For a work typically identified with urgency and propulsion, *On the Road* could be read at a different speed and velocity, one reminiscent of that in Chaucer. Sal Paradise is not just interested in "'balling'" across North America; he, too, shows the ability to sense "the in-between places, in search of people whose relationship with the landscape and environment still shows some authenticity" (Arnds 2020, p. 5). In Denver in spring of 1949: "At dusk I walked. I felt like a speck on the surface of the sad red earth ... At lilac evening I walked with every muscle aching ... strolling in the dark, mysterious streets" (179–80). At a jazz club in San Francisco, Slim Gaillard beats the bongos. He will "suddenly slow down the beat and brood over his bongos with fingertips barely tapping the skin as everybody leans forward breathlessly to hear; you think he'll do this for a minute or so, but he goes right on, for as long as an hour, making an imperceptible little noise with the tips of his fingernails, smaller and smaller all the time till you can't hear it any more and sounds of traffic come in the open door" (176). This is what Kerouac does with his own prose, persistently slowing down so we pay attention to each element. Sal becomes one with insects in Mexico.

> I realized the jungle takes you over and you become it. Lying on the top of the car with my face to the black sky was like lying in a closed trunk on a summer night. For the first time in my life the weather was not something that touched me, that caressed me, froze or sweated me, but became me. The atmosphere and I became the same. Soft infinitesimal showers of

> microscopic bugs fanned down on my face as I slept, and they were extremely pleasant and soothing. The sky was starless, utterly unseen and heavy. I could lie there all night long with my face exposed to the heavens, and it would do me no more harm than a velvet drape drawn over me. The dead bugs mingled with my blood; the live mosquitoes exchanged further portions; I began to tingle all over and to smell of the rank, hot, and rotten jungle, all over from hair and face to feet and toes (294).

Here, Kerouac's slow pilgrimage ecopoetics allows us to pay mindful attention to more-than-human actors, from those on the cosmic scale to microscopic insect life. Kerouac portrays Sal as acting as a temporary tenant of the environment (see Serres 2011, p. 86).

A useful, if venerable, way to explore pilgrimage literature can be found in Viktor Shklovsky's concept of defamiliarization. He suggests "the technique of art is . . . *to increase the difficulty and length of perception* because the process of perception is an aesthetic end in itself and *must be prolonged*" (Shklovksy 1965, p. 12; my emphasis). This lengthening of perception, prolonging our observation, means that we exceed the necessary or utilitarian mode of vision. As their pilgrimage winds to a close, the sacred moment is prolonged and drawn out, lengthening the spiritual impact. Sal writes, "The end of our journey impended . . . We'd made it, a total of nineteen hundred miles from the afternoon yards of Denver to these vast and Biblical areas of the world, and now we were about to reach the end of the road" (299).

## 4. Conclusions: Metatextuality and Literary Encounters

Key to pilgrimage literature extending back into the medieval period is the telling of stories while underway. In *The Canterbury Tales*, each pilgrim is to tell four tales, two from Southwark to Canterbury and two on the return. During the route to Mexico, "Dean and I pounded out plot after plot of books we'd read into Stan" (270). Metatextual allusions parallel *The Canterbury Tales* with *On the Road*. In the medieval work, the figure of the Host cajoles and bullies various figures into telling tales. Works of the historical Geoffrey Chaucer are mentioned in conjunction with his fictional alter ego. In *On the Road*, Sal writes a book that functions as a hook for the reader throughout the novel until it finally becomes published. Each section begins with some reference to the development of his novel. In the opening pages, Sal tells us, Chad "said I was a writer and [Dean] should come to me for advice" (3) and we learn Sal types fast (4). At the start of Part Two, he mentions how he "finished my book" (109). Near the beginning of Part Three, Sal mentions the "book I had finished, which was now accepted by the publishers" (187). At the start of Part Four, "I came into some money from selling my book" (249). By the conclusion of Part Five, the novel has been completed. This self-referentiality by each writer suggests that part of the pilgrimage is a literary journey undertaken through the creation of the written work. Like its medieval counterpart, *On the Road* ends opening outwards: "Did this mean that I should at last go on my pilgrimage on foot on the dark roads around America?" (303).

Kerouac mentions Chaucer, an evident influence, at the outset of shaping *On the Road*. Other influential writers include Shakespeare, whom Kerouac studied with Mark Van Doren, along with Thomas Wolfe, Theodore Dreiser, Goethe, Keats, Proust, and Joyce. In "Shakespeare and the Outsider", Kerouac praises Shakespeare as the greatest writer "in any language in any country anytime in the history of the world [ . . . ]. Compared to him Homer groaned, Dante too [ . . . ] Chaucer sat up in his grave and glanced curiously that away" (Kerouac 1998, p. 85). The parallels between Kerouac and Chaucer highlight elements characteristic of pilgrimage literature rooted in the Middle Ages and continuing into Beat writing.

Was this the only contact between Chaucer and the beat? Other connections include Carol Bergé's "Fragment (A Gift)", in which she mentions a woman named Alisoun, spelled the medieval way (Bergé 1971, pp. 98–99; see, for example, Chaucer's Miller's Tale I.3401 and Wife of Bath's Prologue III.530, 804). Additionally, in the late 1950s, there were events where people read poetry aloud to jazz. One of these moments occurred at the Five Spot Café in Greenwich Village, which Kerouac himself frequented (Amburn 1998, pp. 283, 286). This particular event featured Miles Davis' album, *Bags' Groove*. At one

point in his life, Kerouac told his friend and fellow writer John Clellon Holmes that he planned to write a novel on the jazz scene, including about Miles Davis (Amburn 1998, p. 184). The reader was George Stade, English professor at Columbia University who, with Amiri Baraka—at that time still known as Leroi Jones—used to organize these poetry-jazz fusions. What did George Stade read (Amburn 1998, pp. 286–87)?

> Whan that Aprill with his shoures soote
> The droghte of March hath perced to the roote,
> And bathed every veyne in swich licour
> Of which vertu engendred is the flour;
> Whan Zephirus eek with his sweete breeth
> Inspired hath in every holt and heeth
> The tendre croppes, and the yonge sonne
> Hath in the Ram his half cours yronne. (Chaucer, *The General Prologue*, I.1–8).

**Funding:** This research received no external funding.

**Acknowledgments:** Many thanks to the anonymous reviewers of this essay for their invaluable suggestions and insights. In addition, Wendy Maldonado, Steve Wilson, and Lydia Blanchard offered advice and support. The following people generously answered queries in personal communications: Jacques Barzun; Carl Woodring; Quentin Anderson; George Stade; Jocelyn K. Wilk, University Records, Columbia University; Gary Morrison, Registrar's Office, Columbia University; Barbara Connolly, Horace Mann School; William F. Buckley, Jr.; Jean Ashton, Director, Rare Book and Manuscript Library, Columbia University; Les Meyers, New School University Libraries; Rodney Phillips, Curator, Berg Collection, New York Public Library; John Sampas; Ann Charters; Martha Mayo, Director, University of Massachusetts Lowell, Center for Lowell History.

**Conflicts of Interest:** The author declares no conflict of interest.

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
