# Peer review of "“[A]n Exterior Air of Pilgrimage”: The Resilience of Pilgrimage Ecopoetics and Slow Travel from Chaucer’s The Canterbury Tales to Jack Kerouac’s On the Road"

_humanities, doi:10.3390/h9040117_

Round 1
Reviewer 1 Report
1
Reader Report – “‘[A]n exterior air of pilgrimage’: Slow Travel in Chaucer’s The Canterbury Tales and Jack Kerouac’s On the Road”
Overview: This article analyzes medieval English poet Geoffrey Chaucer’s influence on contemporary American author Jack Kerouac, focusing on how a “pilgrimage ecopoetics” manifests in both authors’ texts. Sections of the article discuss how the theme of pilgrimage permeates The Canterbury Tales and On the Road; how forging (particularly in the sense of deception) is prevalent in both texts; how vernacular speech predominates in both works; and how slow travel is characteristic of the two texts. A final section describes Chaucer’s potential impact on other Beat generation authors and artists, suggesting the resilience of a medieval pilgrimage ecopoetics into contemporary times.
General comments: Overall, the paper was written clearly and was thought-provoking, as I had not previously considered On the Road in relation to The Canterbury Tales. I was especially intrigued by the idea of a pilgrimage ecopoetics rooted in the medieval period but potentially continuing into the Beats’ time and today. In fact, maybe that would be a useful term to include in the paper’s title. Otherwise, my main comments to improve the piece involve the following:
Converting what appears to be a conference paper into an article. The genre conventions for the two are related, but different. For instance, articles typically have more protracted background sections. With the genre conversion and a potential revision in the thesis, sections may be expanded or otherwise modified. Considering both similarities and differences between the texts under discussion. Upon reading the first section, for instance, I thought of several potential differences between the pilgrimages in the two texts: (1) to what extent the pilgrims have definite destinations; (2) what role faith, both in terms of organized religion and spirituality more broadly, plays in the works; and (3) how technology impacts the journeys distinctly. The paper does allude to these points, but they may be worth unpacking more. Perhaps pilgrimage ecopoetics has evolved substantially since the medieval period, although elements of the earlier era are still present in Anglo-American poetry of the last 50-75 years. Clarifying the paper’s thesis and its distinctive intervention in the critical conversation. The first paragraph gestured at a thesis toward the end, but overall the claims there were more topical than argumentative. As suggested above, I like the idea of a pilgrimage ecopoetics at the intersection of religious studies, the environmental humanities, and literary criticism. The challenge is then crystallizing an argument about the connections between Canterbury Tales and On the Road at the nexus of those scholarly fields.
Specific comments (by section, as the draft had no line numbers):
Introductory paragraph: Here, I wasn’t sure if the paragraph was intended to be an abstract or an introduction to the paper. There was also an allusion to a text that readers may not be aware of (Blue Highways). As indicated above, with article genre conventions readers would be more immersed in the literary and scholarly contexts leading up to the paper’s main argument. There is also a typo at the end of the last sentence (a missing period). I noticed a couple other minor grammatical and typographical issues, so it may be worth scouring over the piece again while revising.
2
Section 1: The overall points here were fine, but as with some of the other sections, the section began with a close reading before suggesting the main point. Readers may expect to see a thesis sentence earlier so as to better understand the close reading. For example, I thought Kerouac’s religious background could have been presented before the close readings in order to situate them in a broader context. Otherwise, both here and elsewhere, it may be worth casting a more critical eye on the protagonists. E.g., is Mexico being exoticized problematically in the imaginations of the characters?
Section 2: This section seemed a bit disconnected to me from an ecocritical argument specifically. Also, a citation to Vasvári is missing.
Section 3: Again, leading with stronger thesis sentences would be helpful rather than juxtaposing the two texts and then presenting the main point. Otherwise, it struck me as interesting, though not surprising, that the aesthetic lineage cited at the bottom of page 4 appears to be mostly (or all) male and white. While gender is not necessarily the main focus of this particular article, gender inclusivity (or lack thereof) in the Beat movement is a topic of scholarly attention and may be useful to consider.
Section 4: Here, the pilgrimages under discussion were helpfully distinguished, as suggested earlier. What slow travel entailed in the medieval period versus the early contemporary period may be worth noting. The idea of slow travel also evoked for me the various critiques of life’s pace today, e.g., The Slow Professor. A slower pace of travel arguably fosters more contemplation and interaction (with other people, nature, etc.).
Section 5: The closing paragraphs here may be better adapted into a wider introductory frame for the paper, presenting the larger issue of connections between Chaucer and the Beats before homing in on Chaucer and Kerouac more specifically. Also, could it be said that Chaucer was more a subconscious, though not necessarily minor, influence on Kerouac? I haven’t scrutinized the primary sources enough to say, but there do seem to be important connections between the two and there’s no reason to undersell the argument if a stronger claim can reasonably be made.
Author Response
I’m very grateful to the excellent comments and specific suggestion made by Reviewer 1 (R1). The title is altered to reflect R1’s helpful suggestions in the General comments section. In converting what was a conference paper into an article, I needed to flesh out certain aspects which R1 helped guide me towards. I’ve expanded various things, including the issue of definite destinations by pilgrims; faith itself; and technology as impacting journey. Taking Reviewer 2’s suggestion to weave in more reference to the Asian influence on Kerouac, I feel that this expansion is better textured.
R1 points out that I didn’t have enough guideposts to the reader in the form of direct theses and topic sentences in the different sections. I hope I’ve sufficiently taken care of them.
Concerning specific suggestions:
I both identified Blue Highways and moved it until later in the essay; also added Solnit’s book in order to present a web of connected works associated with the “slow” movement.
I switched the order of paragraphs to address the concern about having Kerouac’s religious background before the close readings. As for Section 2 and its lack of connection to ecocriticism: I integrate in a much reduced form this section into the essay earlier where I discuss Kerouac’s faith and pilgrimage in general. It seems a better placement here. As suggested, I cut Vasvári, edited and reduced this section and moved it to earlier place in order to establish structural similarities between texts.
I’ve reduced the number of sections with fuller introductions and more fully developed thesis for each section.
I developed changes and enhancement of slowness.
I was bolder in enhancing Chaucer’s influence on Kerouac while acknowledging his role as less important than, say Shakespeare. I don’t want to undersell the argument either, so truly appreciate all this reviewer’s suggestions.
Reviewer 2 Report
I find this to be a very engaging and interesting article that presents Kerouac in a way that will be useful to scholars working on Beat literature, travel writing, and place-based environmental literature.
I do find the manuscript in its current form to be rather heavily focused on Kerouac's interest in European literary and intellectual traditions, but he was also, of course, deeply interested in Asian traditions, such as the tradition of the Chinese hermit. I suggest that there be at least a note about the connection between Kerouac and the idea of the hermit/pilgrim in ancient Chinese literature. For instance, the author might refer to Hong Cheng's article "Communication Across Space and Time: Contemporary American Nature Writers and the Ancient Chinese Poet Han-Shan," which appeared in ISLE 13.1 (Winter 2006): 135-45 and which discusses Kerouac (especially The Dharma Bums). This would help to present a slightly more balanced view of Kerouac's engagement with the pilgrimage motif, even if the main emphasis here in on Kerouac and Chaucer.
Author Response
I found the Hong Cheng article fantastic. It filled in a gap I sorely needed in this article. I hoped it helped flesh out the Asian tradition a bit (I also included reference to the Japanese writer, Bashō). A full-scale examination of the Asian-Chaucer-Kerouac links is not the key purpose of this article, but I hope this enhancement of the material inspired by Cheng has improved the article. I’m grateful to this reviewer for this suggestion. And I’ve explore Kerouac’s engagement with pilgrimage more.
Round 2
Reviewer 1 Report
Reader Report – “‘[A]n exterior air of pilgrimage’: The Resilience of Pilgrimage Ecopoetics and Slow Travel from Chaucer’s The Canterbury Tales to Jack Kerouac’s On the Road”
Overall, the draft reflects an improvement over the prior version. I have included feedback by section below. The version I received did not have line numbers, so I’m using section designations instead. Aside from the substantive points noted below, I noticed a few minor typos, so it may be useful to review the draft for grammar and style issues before publication.
Introduction
- Framing: The new draft contains a broader frame for the piece, but more here would be useful before homing in on the article’s focus (Chaucer/Kerouac).
- Critical conversation/why care question: As noted in the prior draft, greater immersion in the critical conversation would be helpful. What larger issues in the relevant scholarly fields are illuminated through the case study?
- Minor observation (doesn’t necessarily need to be included in the revised draft): The first paragraph reminded me of Eliot’s “Tradition and the Individual Talent” re: the fluid nature of tradition (past texts informing our view of present ones and vice versa).
Literary tradition of pilgrimage
- Genre: More background here on the genre of pilgrimage literature may be helpful, particularly conventions and the typical plot trajectory (if there is one). For instance, on page 3 the article mentions the importance of first-person narration in both texts, but is this typical for pilgrimage literature more generally?
- New inclusions:
-
- I’m still unclear about the exact extent of Asian influences on Kerouac’s work; the wording is vague. Are there specific passages supporting the claims here?
- Note that Kerouac’s immediate predecessors, i.e., Anglo-American modernists, did often rely on medieval and Asian texts, so he could be seen as following in their stead.
- Expanding on the connections between being in nature and increased spirituality in Christian and other faith traditions engaged with could be helpful.
-
- Connecting the discussion about pilgrims going astray to ecocritical concepts would be useful.
- Connecting the discussion about pilgrims going astray to ecocritical concepts would be useful.
Vernacularity, sociality, and topopoetics
- Close reading: A more sustained ecocritical reading would be helpful here. For instance, can “natural” composition processes and “natural” uses of language (maybe the vernacular) be considered part of an ecopoetic aesthetic? If so, why and how? The opening lines of the “Pilgrimage poetics” paragraph seem pertinent to this line of inquiry and should potentially be moved up in the section.
- First paragraph: Much of what is said here is true of “great” literature more generally, so it may be useful to explain what is distinctive about the argument.
- “Vernacularity is . . .” paragraph: Meandering could be seen as a more “natural” movement than planned travel. Also, regarding Sal’s wandering, he may be a classicpostmodern subject, or we could see him as a contemporary flâneur but out in the landscape rather than traversing the city like his 19th century counterparts.
- “Nature writers” paragraph: The idea of wilderness as providing both a respite from and potentially higher form of civilization is common in American letters (e.g., Emerson).
Slow travel
- Section title: The section’s main point seems to be to connect slow travel with a mode of heightened perception that keen observers of nature have. Perhaps make that clearer in the section title.
- Chaucer: A discussion of Canterbury Tales was largely absent from this section but should be included since the article is presenting a comparative study.
- “an Indian thing”: This could be seen as a problematic reference, to the extent Kerouac associates Native Americans with an exoticized natural past (like the Mexico references the article also discusses).
Conclusion
Section title: I might include a colon and reference to the main subject(s) of the section – perhaps metatextuality and/or literary traditions.
Author Response
To the Reviewer: Thank you so much for your specific suggestions. They really help while editing! I've pasted your comments below with my changes in red. Again, Thank you!
I’ve moved the footnotes to the text itself or, in the case of acknowledging those who supported the project, prior to the Works Cited.
Overall, the draft reflects an improvement over the prior version. I have included feedback by section below. The version I received did not have line numbers, so I’m using section designations instead. Aside from the substantive points noted below, I noticed a few minor typos, so it may be useful to review the draft for grammar and style issues before publication.
Introduction
- Framing: The new draft contains a broader frame for the piece, but more here would be useful before homing in on the article’s focus (Chaucer/Kerouac).
I’ve added a bit more general context concerning authors who are responding to previous writers.
- Critical conversation/why care question: As noted in the prior draft, greater immersion in the critical conversation would be helpful. What larger issues in the relevant scholarly fields are illuminated through the case study?
I’ve added further context here to expand as you suggest.
- Minor observation (doesn’t necessarily need to be included in the revised draft): The first paragraph reminded me of Eliot’s “Tradition and the Individual Talent” re: the fluid nature of tradition (past texts informing our view of present ones and vice versa). Great point! I’ve added Eliot to enhance the Chaucer-Kerouac links
Literary tradition of pilgrimage
- Genre: More background here on the genre of pilgrimage literature may be helpful, particularly conventions and the typical plot trajectory (if there is one). For instance, on page 3 the article mentions the importance of first-person narration in both texts, but is this typical for pilgrimage literature more generally?
I’ve added a bit more information here. I hope it’s enough.
- New inclusions:
- I’m still unclear about the exact extent of Asian influences on Kerouac’s work; the wording is vague. Are there specific passages supporting the claims here?
I’ve included more from the Dharma Bums to support this claim. I’ve cut the references to Bashō to streamline the argument.
- Note that Kerouac’s immediate predecessors, i.e., Anglo-American modernists, did often rely on medieval and Asian texts, so he could be seen as following in their stead.
- Expanding on the connections between being in nature and increased spirituality in Christian and other faith traditions engaged with could be helpful.
Connecting the discussion about pilgrims going astray to ecocritical concepts would be useful.
I’ve added a brief explanation on how straying can prevent the pilgrim-actor from feeling as though he can master the world around him.
Vernacularity, sociality, and topopoetics
- Close reading: A more sustained ecocritical reading would be helpful here. For instance, can “natural” composition processes and “natural” uses of language (maybe the vernacular) be considered part of an ecopoetic aesthetic? If so, why and how? The opening lines of the “Pilgrimage poetics” paragraph seem pertinent to this line of inquiry and should potentially be moved up in the section.
I’ve moved some of the “pilgrimage poetics” paragraph up to the beginning of this section as suggested. I feel this does address some issues in this section here. I’ve also added a phrase to clarify this.
- First paragraph: Much of what is said here is true of “great” literature more generally, so it may be useful to explain what is distinctive about the argument.
I’ve moved the final sentence from the “Pilgrimage poetics” paragraph here which seems to answer this question.
- “Vernacularity is . . .” paragraph: Meandering could be seen as a more “natural” movement than planned travel. I’ve added a sentence here. Also, regarding Sal’s wandering, he may be a classicpostmodern subject, or we could see him as a contemporary flâneur but out in the landscape rather than traversing the city like his 19th century counterparts.
See the added reference to Gersdorf; thank you for the flaneur allusion!
- “Nature writers” paragraph: The idea of wilderness as providing both a respite from and potentially higher form of civilization is common in American letters (e.g., Emerson). I’ve added this fine point.
Slow travel
- Section title: The section’s main point seems to be to connect slow travel with a mode of heightened perception that keen observers of nature have. Perhaps make that clearer in the section title.
I’ve added a further half title
- Chaucer: A discussion of Canterbury Tales was largely absent from this section but should be included since the article is presenting a comparative study.
I’ve added a half paragraph to indicate a couple of examples of slow travel in Chaucer.
- “an Indian thing”: This could be seen as a problematic reference, to the extent Kerouac associates Native Americans with an exoticized natural past (like the Mexico references the article also discusses).
I’ve cut this reference to streamline the discussion. To go into the problematics of “Indian” here could entail a digression the article cannot afford to take.
Conclusion
Section title: I might include a colon and reference to the main subject(s) of the section – perhaps metatextuality and/or literary traditions.
I’ve taken your excellent suggestion here. “Conclusion: Metatextuality and Literary Encounters” not only explicitly lets the reader know metatextuality will be discussed, but ties the conclusion of the article back to the beginning and Hong Cheng reference to intertextual ‘‘encounters, perhaps conversations.”